# Guiding Language Model Reasoning with Planning Tokens

**Xinyi Wang**[1]*, **Lucas Caccia**[2], **Oleksiy Ostapenko**[2,3,4], **Xingdi Yuan**[2],
**William Yang Wang**[1], **Alessandro Sordoni**[2,3,4]
[1]University of California, Santa Barbara    [2]Microsoft Research
[3]Mila — Quebec AI Institute    [4]Université de Montréal
xinyi_wang@ucsb.edu, william@cs.ucsb.edu, {eric.yuan, alsordon}@microsoft.com

## Abstract

Large language models (LLMs) have recently attracted considerable interest for their ability to perform complex reasoning tasks, such as chain-of-thought (CoT) reasoning. However, most of the existing approaches to enhance this ability rely heavily on data-driven methods, while neglecting the structural aspects of the model's reasoning capacity. To encourage a more structural generation of CoT steps, we propose a hierarchical generation scheme: we let the LM generate a *planning token* at the start of each reasoning step, intuitively serving as a high-level plan of the current step, and add their embeddings to the model parameters. Our approach requires a negligible increase in trainable parameters (0.001%) and can be applied through either full fine-tuning or a more parameter-efficient scheme. We demonstrate our method's effectiveness by applying it to three different LLMs, showing notable accuracy improvements across three math word problem datasets and one multihop QA dataset with respect to standard fine-tuning baselines.[1]

## 1 Introduction

The great potential of solving complex reasoning problems, including world knowledge reasoning (Hendrycks et al., 2020; Suzgun et al., 2022), logical reasoning (Pan et al., 2023), and math reasoning (Cobbe et al., 2021; Hendrycks et al., 2021b), using pre-trained large language models (LLMs) (Touvron et al., 2023a;b; Brown et al., 2020) has drawn much attention recently. A popular and effective paradigm of reasoning with LMs is chain-of-thought (CoT) reasoning (Wei et al., 2022; Wang et al., 2022). In CoT the LMs are required to generate both the reasoning steps and answer for a given problem. Recent research has recognized the importance of CoTs for complex reasoning problems. Multiple works focus on augmenting high-quality alternative CoTs in training data. For example, Yue et al. (2023) fine-tune LLMs on multiple math datasets with CoT and program-of-thought (PoT) solutions. Yuan et al. (2023) applies rejection sampling on the LLM samples. Other works elicit reasonings from exogenous resources, such as more capable LLMs, i.e. GPT-4 (Mukherjee et al., 2023; Luo et al., 2023). Although these methods are shown to be highly effective, the fundamental problem of how to effectively guide an LM to generate more useful CoT is not directly tackled by these data-augmentation-based methods.

We aim to propose a lightweight training method to fundamentally improve the effectiveness of the generated CoTs. While current methods usually directly generate the complete CoT solution, we hypothesize that a hierarchical generation of the CoT steps will benefit the overall quality of the solution. More specifically, we introduce *planning tokens* at the beginning of each CoT step, which are special tokens that encode a high-level solution plan of the current CoT step. We consider three ways of inferring the *planning tokens*: heuristic assignment, clustering reasoning states, and latent variable learning with VAE (Kingma & Welling, 2013). At training time, we first assign *planning tokens* to each CoT step and

---

*This work was done during an internship at Microsoft Research, Montreal.
[1]We open source the code at https://github.com/WANGXinyiLinda/planning_tokens.

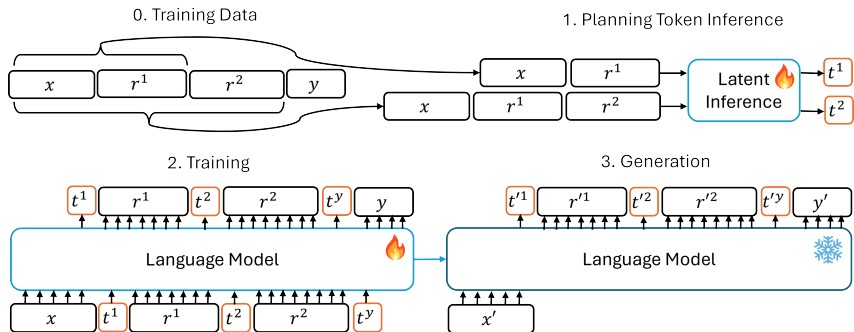

Figure 1: An overview of our method: Given an input question $x$ and its CoT steps $r^1, ..., r^i$, we train a planning token inference model to infer the planning tokens $t^i$. Subsequently, we fine-tune an LM on data augmented with the inferred planning tokens. Given a new question $x'$, the trained LM can infer planning tokens $t'^i$, CoT steps $r'^i$, and final answer $y'$. Fire and snowflake denote trainable and frozen models, respectively. Note that $x, r^i, t^i, y$ can all include **multiple tokens**, we depict them as single blocks/variables for simplicity.

then do regular supervised fine-tuning. At inference time, the LM will be able to generate the corresponding *planning tokens* first, before generating each CoT step. In practice, the planning tokens' embeddings add negligible additional learnable parameters to the LLMs ($< 0.001\%$). This makes our method integrate well with parameter-efficient fine-tuning methods like LoRA (Hu et al., 2021).

While the idea of adding new tokens to the generative LM's vocabulary and then training the associated embeddings has been explored before (Li & Liang, 2021; Lester et al., 2021), the function and effect of our proposed planning tokens are significantly different from the previous works. Our *planning tokens* are designed to increase and guide the reasoning ability of LM fine-tuned with other supervised fine-tuning methods, instead of acting as a parameter-efficient fine-tuning method (Li & Liang, 2021; Lester et al., 2021) on its own. The fact that *planning tokens* are specialized within one task and can be generated and inferred by LMs like regular tokens also sets us apart from the previous soft token tuning based methods. For example, Qin & Eisner (2021) treat the soft tokens as special indicators for specific relations, and Goyal et al. (2023) fixed the position of the pause token to right after the prompt and serves only as extended computation before generation.

We perform experiments on three math word problem (MWP) datasets (GSM8K (Cobbe et al., 2021), AQUA (Ling et al., 2017), and MATH (Hendrycks et al., 2021a)), and one multihop QA dataset (StrategyQA (Geva et al., 2021)). The experiment results in Section 3.1 show that by adding planning tokens at training time, we are able to improve upon the baseline without planning tokens by 3.3% accuracy points on average over three pre-trained language models (Phi 1.5 (Gunasekar et al., 2023), 7B Llama 2 (Touvron et al., 2023b), and 13B Llama 2) and three MWP datasets. We also show that the gains of our method do not solely come from the augmented computational space induced by increasing the length of the reasoning with planning tokens but also from the specialization induced by the planning tokens. In Section 3.2, from a detailed error analysis of the generated CoT solutions, we found that our method performs notably better on longer CoT solutions. The error type analysis also suggests that our method can better refer to the previous information and increase the computing capacity of the model. We also investigate how LLMs utilize the planning tokens by looking into the attention weights, which suggest that planning tokens can carry information throughout the whole sequence.

## 2 Method

**Setup** We assume that we have a dataset $\mathcal{D}$ composed of triples $\{(x, r, y)\}$, where $x$ is a problem to solve, $r$ is the ground-truth reasoning associated with the problem and $y$ is the

final answer. The assumption that the ground-truth reasoning is included in the dataset might be circumvented by either generating CoTs with rejection sampling from a base LM (Yuan et al., 2023) or by creating reasoning paths from a larger model (Luo et al., 2023; Mukherjee et al., 2023). We fine-tune a base LLM to predict the concatenation of reasoning and answers given the inputs, i.e.:

$$\arg\max_\theta \sum_\mathcal{D} \log p(r, y|x; \theta),$$

where we assume that $x, r, y$ are properly tokenized and $\theta$ is the trainable parameter of the pre-trained LLM. To provide more high-level guidance to the LLM at fine-tuning time with reasoning data, we assume that each reasoning $r$ is the realization of some higher-level planning process, which is *latent*, thus unobserved in the dataset. To model this assumption, we define by *reasoning step* a sub-sequence of tokens appearing in the reasoning. We create reasoning steps by splitting the reasoning with a delimiter token, which might be dataset-specific, but for simplicity, we fix it to be \n for all datasets.

Let $r = r^1, \ldots, r^S$, where $S$ is the number of steps in the reasoning. We assume that each reasoning step $r^i$ is generated conditional on the previous reasoning steps and a discrete latent planning variable $t^i$ taking values into $1 \ldots P$. Given that the planning variables value between 1 and $P$, they can be considered as condensed summaries of each reasoning step and can therefore provide useful guidance while producing the reasoning $r$. We also assume the answer $y$ is generated with a special planning variable $t^y$ which has always a value equal to $P$ for each example. Without loss of generality, we assume that only $t^y$ can take value $P$, i.e. $p(t^i = P) = 0$ for each $t^i, i \neq y$.

If the planning variables were indeed observed, the complete data $\mathcal{D}^c$ would be composed of tuples $(x, t^1, r^1, \ldots, t^S, r^S, t^y, y)$. Let us assume for the moment that the complete dataset is available to us. In order for the dataset to be processed by an LLM, we verbalize each planning variable $t^i$ with a (or possibly multiple) *planning token*(s). Here we abuse the notation $t^i$ for this (set of) *planning token*(s) for simplicity. In practice, this means that we extend the vocabulary of the LLM tokenizer with $P$ (or more) new tokens of our choice, and modify accordingly the input and output embedding matrix of the LLM. The embeddings of the new tokens will be additional training parameters for the LLM, which add a negligible parameter overhead over the base model.

With this modification in mind, the complete data $\mathcal{D}^c$ can be readily used to train an LLM by maximizing the complete data log-likelihood:

$$\arg\max_{\theta^+} \sum_{\mathcal{D}^c} \log p(t^1, r^1, \ldots, t^S, r^S, t^y, y|x; \theta^+),$$

by $\theta^+$ we denote the original LLM parameters comprising the embeddings of the planning tokens. From the equation above, we can see that the LLM will be trained to first generate the high-level planning token for each reasoning step, and then generate the detailed reasoning step. This can help LLMs to generate more consistent and controlled reasoning chains.

## 2.1 Planning Tokens Inference

We consider three different ways of inferring the planning variable from the given CoT training data: **Arithmetic**, **K-Means**, and **SQ-VAE**. In practice, this inference step corresponds to assigning the instantiated planing tokens to the reasoning steps prior to fine-tuning the LM and learning the embeddings of the planing tokens.

**Arithmetic**   For math word problems, it is natural to consider the basic arithmetic operation contained in each reasoning step $r^i$ as the plan token $t^i$ similar to  Zhang et al. (2023); Qin & Eisner (2021). This yields four planning tokens (+, -, ×, ÷), which can be easily extracted from each $r^i$. If more than one operator is found, then we use as many planning tokens as the number of operations found, e.g. <+><*>.

This approach has several drawbacks. First, some operations, such as doubling a quantity, can be implemented with more than one operator (e.g. + and ×). Thus, this arithmetic

partitioning can fail to capture the similarity between these reasoning steps. Second, for some more complex datasets such as MATH, where the reasoning goes beyond basic operators, such heuristics might not be simple to design if not provided by the dataset. Finally and most importantly, such a heuristic does not apply to non-math datasets like multihop QA datasets. To relax these shortcomings, we turn to inference planning tokens from the embeddings of CoT steps.

We first gather reasoning steps $r^i$ for all examples in our dataset, then apply an inference algorithm on the representations $\phi(r^i), \forall r^i \in \mathcal{D}$. We obtain $\phi(r^i)$ by using an LLM to encode the whole text sequence of an example $(x, y)$, and then averaging the last hidden layer outputs of the corresponding tokens in $r^i$. In this way, we aggregate both the question and previous steps' information in the contextualized representation $\phi(r^i)$.

**K-Means**   We run an off-the-shelf K-Means implementations on the set of representations $\phi(r^i)$ for all reasoning steps in the dataset $\mathcal{D}$. Each planning variable $t^i$ is then assigned the index of the centroid closest to $r^i$.

**SQ-VAE**   In this approach, our aim is to learn a non-linear transformation of each representation $\phi(r^i)$ that better captures the meaningful directions of variation of the representations for the dataset $\mathcal{D}$ under consideration. Variational Autoencoders (VAEs) (Kingma & Welling, 2014) offer a probabilistic approach to learning such non-linear latent representations of data. In their original formulations, VAEs learn a Gaussian-distributed latent space of representations. To induce a discrete structure in the latent space, we follow Miao et al. (2017), and use a "Gaussian-softmax" parameterization, which soft-quantizes the latent representations before reconstructing the input data. In particular, our VAE maximizes the following objective:

$$\mathcal{L}(enc, dec, \Phi) = \mathbb{E}_{enc(z^i|\phi(r^i))}[|||dec(\phi(r^i)|\bar{z}) - \phi(r^i)||_2] - \text{KL}(enc(z^i|\phi(r^i)), \mathcal{N}(0, I)),$$

$$\bar{z} = \Phi q, \quad q = softmax(z^i)$$

where *enc* and *dec* are the encoder and the decoder neural networks that transform the steps representation into the non-linear manifold and back into the original representation space respectively (the encoder parameterizes mean and log-variance of a Gaussian distribution); $q$ is the soft-quantization of the Gaussian-distributed representation $z^i$, which corresponds to the distribution over clusters. $\Phi$ is a learnable matrix with $P - 1$ rows (the last value is reserved for the answer planning variable) representing the cluster centroids. Therefore the model tries to reconstruct the input representation given a soft-combination of cluster centroids. With this model, after training, we obtain the assignment for the planning variables as $t^i = \arg\max softmax(enc(\phi(r^i)))$, where $enc(\phi(r^i))$ denotes the mean of the Gaussian distribution parameterized by *enc*.

### 2.2 Planning Tokens Parametrization

Each planning variable may be verbalized by one or more planning tokens. Moreover, we discussed two orthogonal intuitive effects that might be achieved by augmenting the dataset with planning tokens. The first is to augment the computational capacity of the model by providing additional "scratch" space to predict the next reasoning step; the second is the specialization induced by information-bearing planning tokens. To be able to disentangle both effects in our model, we introduce two hyper-parameters in our approach: *n_prefix* and *n_special*. Each planning variable can be verbalized using a variable number of both generic *prefix* planning tokens and *special* planning tokens. A planning token annotated example with *n_prefix* = *n_special* = 3 is shown below:

Question: Chenny is 10 years old. Alyana is 4 years younger than Chenny. How old is Anne if she is 2 years older than Alyana?

<prefix_0><prefix_1><prefix_2><kmeans1_0><kmeans1_1><kmeans1_2> Alyana is 10 - 4 = <<10-4=6>>6 years old.

<prefix_0><prefix_1><prefix_2><kmeans3_0><kmeans3_1><kmeans3_2> So, Anne is 6 + 2 = <<6+2=8>>8 years old.

<prefix_0><prefix_1><prefix_2><answer_0><answer_1><answer_2> The answer is: 8

| Model | Method | #clusters | #trainable | GSM8K | MATH | AQUA | S-QA | Avg |
|---|---|---|---|---|---|---|---|---|
| Phi 1.5 | Full-FT | 0 | 100% | 12.5 | 1.3 | 27.2 | - | 13.5 |
| (1.3B) | + General | 1 | 100% | 15.4 | 2.0 | 35.4 | - | 17.6 |
| | + **Arithmetic** | 4 | 100% | 15.0 | 2.3 | 33.1 | - | 16.8 |
| | + **K-Means** | 5 | 100% | 14.5 | 2.7 | **36.5** | - | 17.7 |
| | + **SQ-VAE** | 5 | 100% | **15.8** | **3.3** | 34.3 | - | **17.8** |
| Llama2 | LoRA | 0 | 0.343% | 38.2 | 6.5 | 36.6 | 58.4 | 34.9 |
| (7B) | + Pause | 1 | 0.344% | 37.2 | 6.7 | 36.2 | 58.0 | 34.5 |
| | + General | 1 | 0.344% | 38.5 | 6.7 | 37.8 | 59.1 | 35.5 |
| | + **Arithmetic** | 4 | 0.344% | 39.5 | 5.6 | 38.2 | - | - |
| | + **K-Means** | 5 | 0.344% | 39.1 | 6.7 | 40.5 | 62.2 | 37.1 |
| | + **SQ-VAE** | 5 | 0.344% | **40.0** | **7.0** | **41.3** | **62.8** | **37.8** |
| Llama2 | LoRA | 0 | 0.279% | 44.6 | 7.2 | 41.3 | 69.5 | 40.7 |
| (13B) | + General | 1 | 0.280% | 47.9 | 7.9 | 42.5 | 70.3 | 42.2 |
| | + **Arithmetic** | 4 | 0.280% | 41.9 | 4.6 | 35.8 | - | - |
| | + **K-Means** | 5 | 0.280% | 49.6 | 8.4 | **44.1** | 71.5 | 43.4 |
| | + **SQ-VAE** | 5 | 0.280% | **50.6** | **8.5** | 43.9 | **72.4** | **43.9** |

Table 1: Testing accuracy of fine-tuned language models on different math word datasets. We set the number of planning tokens ($n\_prefix+n\_special$) to 6.

Here <kmeans1_0>, <kmeans1_1>, and <kmeans1_2> are the three planning tokens induced by the first K-Means cluster.

## 3 Experiments

**Datasets** We conduct experiments on four datasets. The Grade School Math dataset (**GSM8K**) (Cobbe et al., 2021) contains 8.5K examples of linguistically diverse grade school math world problems. The **MATH** dataset (Hendrycks et al., 2021a) is a collection of 12.5K challenging competition mathematics problems formatted in latex notation. The **AQUA**-RAT dataset (Ling et al., 2017) contains 100K samples of mathematical problems, along with sequences of human-readable mathematical expressions in natural language. **StrategyQA** contains 3K multi-hop questions annotated with decomposed single-hop questions, which we used as the Chain-of-thought (CoT) path of the question.

**Base LLMs** Our empirical analysis uses several decoder-only architectures of varying sizes. We use the 7B and 13B variants of Llama2 (Touvron et al., 2023b), both trained over 2 trillion tokens from publicly accessible data sources. We also experiment with Phi-1.5 (Gunasekar et al., 2023), a 1.3B parameter model trained on a mixture of textbook-quality code data, and additional synthetically generated textbook and exercise data.

**Baselines** First, we compare against the vanilla baselines of full-fine-tuning (**Full-FT**) for Phi-1.5 and **LoRA** fine-tuning for Llama-2. Then, we compare to a baseline that combines plain soft token tuning (**General**) with the base fine-tuning method (**Full-FT/LoRA**). For this baseline, we prepend the same prefix tokens to each CoT step. i.e. $P = 1$.

Note that the **General** baseline can be viewed as prompt-tuning (Lester et al., 2021) + the base fine-tuning method, when the prefix tokens are only prepended to the first CoT step. To make prompt-tuning more comparable to our planning token methods, we enhance the original prompt-tuning by also prepending prefix tokens to other CoT steps.

We also tried the prompt-tuning (Lester et al., 2021) and prefix-tuning (Li & Liang, 2021) only parameter efficient tuning baseline. We found that they are not suitable for complex CoT reasoning tasks in our setting, as their capacity is limited when using a reasonable number (similar to our method) of tunable tokens. The inference time and memory complexity become unreasonable when we try to scale the trainable parameters to be similar to LoRA (3k tunable tokens). Detailed results can be found in the Appendix.

**Our Methods** Our methods are denoted with the three different planning types: **Arithmetic**, **K-Means** and **SQ-VAE**.

| Plan Type | Number of Clusters $P$ | | | | |
|---|---|---|---|---|---|
| | 1 | 3 | 5 | 7 | 10 |
| General | 38.5 | - | - | - | - |
| K-Means | - | 37.9 | 39.1 | 39.9 | 36.7 |
| SQ-VAE | - | 39.6 | 40.0 | 39.4 | 38.9 |

| Plan Type | Number of Plan Tokens | | |
|---|---|---|---|
| | 2 | 6 | 10 |
| General | 37.9 | 38.5 | 38.3 |
| Arithmetic | 39.5 | 38.9 | 38.5 |
| K-Means | 38.9 | 39.1 | 38.1 |
| SQ-VAE | 40.0 | 38.8 | 38.2 |

Table 2: Impact of varying the number of clusters (left) and varying the number of planning tokens (right) (*n_prefix+n_special*). We set *n_prefix = n_special*. For general, we match computation by adding prefix tokens.

## 3.1 Results

We present our main results in Table 1. Generally, we observe that for all three datasets considered and all the model sizes, the best-performing approach leverages planning tokens. We note that, across scales, **Full-FT + General** and **LoRA + General** improves over vanilla fine-tuning (**Full-FT** or **LoRA**), echoing our understanding from Chi et al. (2023) and Feng et al. (2023) that adding additional tokens before each reasoning step increase the compute capacity of the LM and results in better performance. For the three datasets, **Arithmetic** does not consistently outperform **General** pointing to the fact that hand-designed heuristics might not be optimal and the gain observed over **Full-FT/LoRA** may due to additional allowed computation space. However, the other two embedding-based planning type inference methods, **K-Means** and **SQ-VAE**, consistently outperform both **General** and **Arithmetic**, pointing to the importance of using machine-learned planning tokens specialization. Over the 7B version of Llama2, all methods with planning tokens provide consistent gains over the baselines of up to 1.7 points, a relative increase of 6%. Our results over the Llama2 13B tell a similar story, where again the best-performing method across all datasets uses planning tokens. We show a 1.5 gain in average accuracy (4.5 % relative gain). Among these two approaches, SQ-VAE seems to be a more reliable approach, as is it always the best-performing method on average. This can be understood as the non-linear latent inference method works slightly better than the linear one. More expressive latent inference methods might be needed to obtain a larger performance improvement.

**Ablation** We show ablation studies with respect to the number of clusters $P$ and the number of planning tokens used in total. A larger $P$ means we classify reasoning steps into more fine-grained planning types. Note that $P = 1$ means we always use the same planning tokens for all steps which is the same as the **General** baseline. From Table 2 (left), We observe that the performance of both **K-Means** and **SQ-VAE** first goes up and then goes down when the number of clusters increases. This likely reflects that our planning token inference models are not able to produce too fine-grained reasoning type. From Table 2 (right), we observe that it is not necessary to assign too many tokens to a planning type. We suspect that too many planning tokens will result in a longer sequence which negatively affects the language model's reasoning ability.

## 3.2 Analysis

**Errors by Reasoning Length** In Figure 2, we show the accuracy by the complexity of the test example as measured by reasoning length on AQUA and GSM8K. For the two datasets, the introduction of planning tokens improves long reasoning performance, thus hinting that the planning tokens provide better control over long reasoning chains. For GSM8K, planning tokens seem to underperform the baseline for short reasoning chains. While both **Arithmetic** and **SQ-VAE** improve on long reasonings, **SQ-VAE** improves more on average. In future work, we can assign position-aware planning tokens for different steps, to enhance the long reasoning improvement.

**Error Taxonomy** After inspecting the generated chain-of-thoughts solutions from all three datasets, we propose to classify the errors into the following five categories:

- **Misunderstanding of question**: generated solution does not answer the given question.

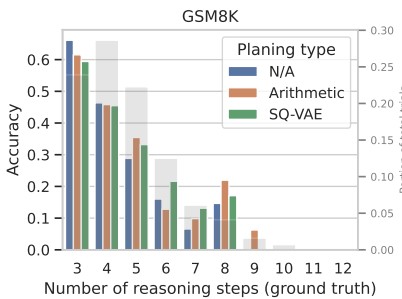 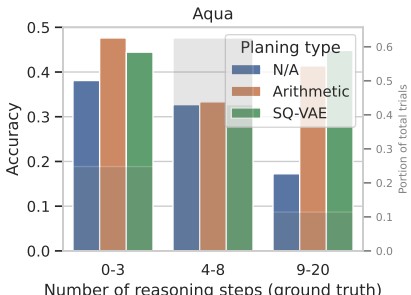

Figure 2: Accuracy on GSM8K (**left**) and Aqua (**right**) on test examples by their number of ground-truth reasoning steps. SQ-VAE consistently increases performance for test examples that require more steps of reasoning to be solved.

| Error type | Misunderstanding of question | | Computation errors | | Inaccurate extraction of question information | | Wrong application of math knowledge | | Wrong logic | |
|---|---|---|---|---|---|---|---|---|---|---|
| Example | Question: ...How many cups of feed does she need to give her chickens in the final meal of the day...
...then each chicken gets 40/20 = <<40/20=2>>2 cups of feed per chicken. The answer is: 2 | | ...7 + 3301x = 3371x... | | Question: ...He spends the next half-hour driving at a speed of 30mph...
...He drove 2 hours at 30mph so he traveled 2*30=<<2*30=60>>60 miles... | | ...The number of feet the plane is from the ground is the target of a geometric sequence... | | ...$2ab = 12 = 2^3 \cdot b^2$... | |
| Dataset | LORA | SQ-VAE | LORA | SQ-VAE | LORA | SQ-VAE | LORA | SQ-VAE | LORA | SQ-VAE |
| GSM8K | 4.1% | 4.1% | 8.0% | **6.1%** | 34.5% | **33.0%** | 0.2% | 0.5% | 16.2% | **14.9%** |
| MATH | 21.0% | 23.5% | 8.1% | **5.0%** | 27.3% | **24.7%** | 20.3% | **18.2%** | 23.1% | 23.0% |
| AQUA | 8.3% | 8.7% | 12.6% | **9.4%** | 25.6% | **23.2%** | 2.8% | 2.0% | 19.5% | 20.1% |

Figure 3: Examples of different error types generated by 7B Llama 2. The frequency of each error type generated by the **LoRA** baseline and our LoRA + **SQ-VAE** planning method on test sets are predicted by GPT4 and then manually verified.

- **Computation errors**: errors in evaluating equations.

- **Inaccurate extraction of question information**: inaccurate references to the information provided in the question.

- **Wrong application of math knowledge**: equations, concepts, or theorems used in the generated solution are not suitable for the question.

- **Wrong logic/unreasonable step**: generation does not follow common logic.

We prompt GPT4 to obtain the error type of each generated example, and then manually verify whether the predicted error type matches the explanation. In Figure 3, the **SQ-VAE** error frequencies significantly lower than the **N/A** baseline are in bold font. As we can see, across three datasets, our method consistently improves on the computation errors and the inaccurate extraction of question information. The lower computation errors indicate that our inferred planning tokens either increase the computation capacity of the model or contain compute-related information. Our method being able to make more faithful reference to the information in the questions indicates planning tokens are helpful for long-range control of generations, which echoes the observation and assumptions in the previous sections. The differences in other error categories are either insignificant or inconsistent over datasets. Note that GSM8K and AQUA have a similar distribution over the error types, while MATH has significantly more misunderstanding of questions and wrong application of math knowledge. This is understandable as MATH is significantly more difficult as it is from American Mathematics Competitions (AMC). These questions are less obvious to approach and require applications of non-trivial math theorems.

**Attention on planning tokens** To better understand how LLMs make use of the planning tokens, we inspect the attention by first computing an overall average of the attention weights assigned to the planning tokens versus the normal tokens. As shown in Table 3, we found that **K-Means** and **SQ-VAE** planning tokens are assigned much higher attention weights than **General** and **Arithmetic** planning tokens, when comparing to normal tokens.

|  | General | Arithmetic | K-means | SQ-VAE |
|---|---|---|---|---|
| Normal token | 0.0045 | 0.0039 | 0.0039 | 0.0043 |
| Planning token | 0.0009 | 0.0012 | **0.0071** | **0.0070** |

Table 3: Averaged attention on planning tokens v.s. normal tokens across all layers, heads, and previous tokens on the test set of GSM8K, with Llama 2 (7B) model.

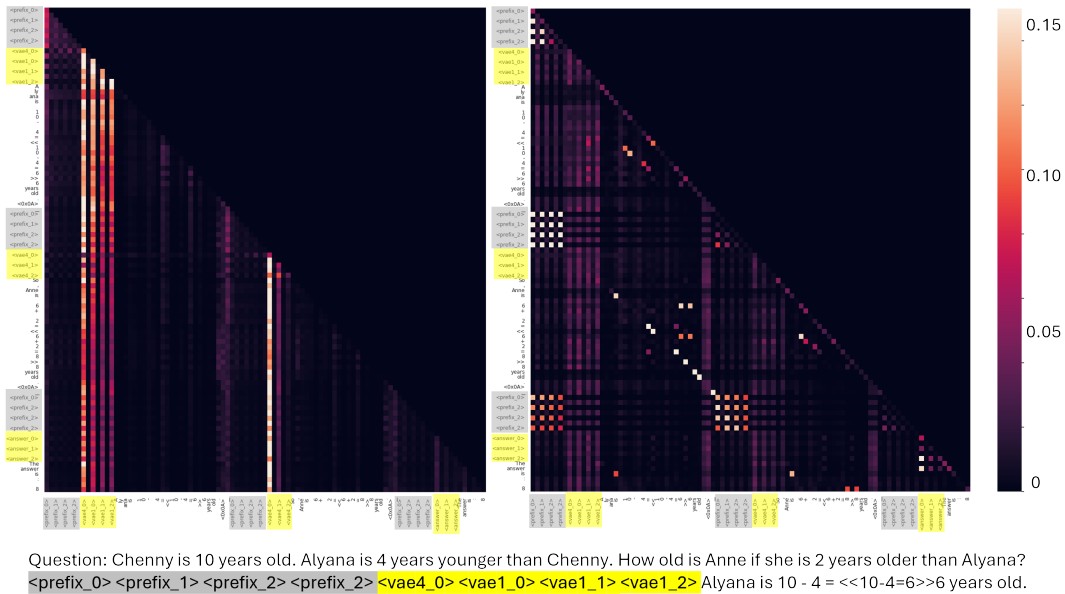

Question: Chenny is 10 years old. Alyana is 4 years younger than Chenny. How old is Anne if she is 2 years older than Alyana?
<prefix_0> <prefix_1> <prefix_2> <prefix_2> <vae4_0> <vae1_0> <vae1_1> <vae1_2> Alyana is 10 - 4 = <<10-4=6>>6 years old.
<prefix_0> <prefix_1> <prefix_2> <prefix_2> <vae4_0> <vae4_1> <vae4_2> So, Anne is 6 + 2 = <<6+2=8>>8 years old.
<prefix_0> <prefix_2> <prefix_2> <prefix_2> <answer_0> <answer_1> <answer_2> The answer is: 8

Figure 4: Two attention heads activated by planning tokens, with a test example generated by Llama 2 (7B) LoRA fine-tuned with **SQ-VAE** planning tokens. Question tokens are omitted in the attention heat map but included in attention computation. A lighter color means a larger attention weight. Best viewed zoomed in.

This implies that the inferred planning tokens might be more helpful for the generation, which is consistent with the main results performance in Table 1.

While the raw attention weight itself might be a debatable way of understanding the token importance, the attention pattern still serves as a valid way of understanding how the Transformer works. Similar to Olsson et al. (2022), we identify attention heads that have strong patterns corresponding to the planning tokens as shown in Figure 4, and deduct how language models make use of the planning tokens from the patterns.

More specifically, we compute the average attention weight difference between planning tokens and normal tokens from different attention heads, and visualize the attention heads with the largest difference, as they are likely to strongly correspond to the utilization of planning tokens. We visualize two such attention heads on a CoT generated with **SQ-VAE** planning tokens in Figure 4. The attention head shown on the left strongly attend to the specialized planning tokens throughout the CoT sequence. The attention head shown on the right shows that the general planning tokens are only attended by themselves, which might imply that the general planning tokens only serve as indicators of the beginning of a CoT step, instead of carrying information to the whole sequence.

**Distinguishability of the Induced Clusterings** We investigate whether **SQ-VAE** learns better planning types than **K-Means** via a probing task (Alain & Bengio, 2017). Concretely, we collect a dataset consisting of CoT steps and their planning token labels obtained from **K-Means** and **SQ-VAE**. We train a simple model (either a logistic regression or a shallow neural network) to learn the mapping from a sentence to its planning token label. We

hypothesize that the better planning type should facilitate easier learning dynamics for simple models because the class boundaries should be easily accessible from the data.

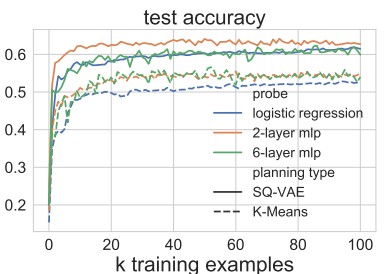

Figure 5: Testing accuracy of the probes on the sentence classification task.

As a case study, we study the 5-cluster setting on the GSM8K dataset. Given a planning type, we randomly sample 2000 sentences from each of the five categories. We leverage a pre-trained text encoder[2] to convert sentences into vector representations. Taking the text encodings as input, the probe network (logistic regression, 2-layer MLP, or 6-layer MLP) performs a 5-way classification.

We show test accuracy curves in Figure 5. We observe a clear gap between **SQ-VAE** (solid lines) and **K-Means** (dashed lines) in the curves. This reflects our main results reported above, where **SQ-VAE** produces a better overall downstream task performance.

## 4    Related Work

**Trainable new tokens** The idea of adding new tokens with trainable embeddings on the input side of a generative LM has been proposed before (Li & Liang, 2021; Lester et al., 2021). The most common way of adding new tokens is to insert them at a fix position in the prompt given to LMs. By only training the embeddings of these new tokens, these methods act as parameter-efficient fine-tuning techniques to adapt LMs to specific task (Qin & Eisner, 2021; Li & Liang, 2021; Lester et al., 2021). Our planning tokens are not intended to serve as a parameter-efficient fine-tuning method. Instead, our method creates a small parameter overhead to the base fine-tuning method and serves as guidance to LM's reasoning process. Qin & Eisner (2021) uses a mixture of soft tokens to model a relation, which tries to specialize within the relation extraction task. However, we do not explicitly compute the mixture probability across all token types as in Qin & Eisner (2021), which can be untractable for CoT generation. Instead, at each step, we let the LM choose one planning token type.

Another line of work prepend newly added tokens as memory to transformers Burtsev et al. (2020); Bulatov et al. (2022); Darcet et al. (2023), which echos our understanding that increasing sequence length can increase the capacity of the Transformer. Goyal et al. (2023) proposes to append a new pause token after the prompt so that the LM can have more computation budget before starting the actual generation. This shares the same idea with the token as a memory line of work. On the contrary, Mu et al. (2023) proposes to append a new gist token after the prompt so that the following generation will only attend to this gist token for compression purposes. Our **General** baseline can be viewed as an enhanced version of Goyal et al. (2023). Instead of only adding new tokens after the prompt, we also add new tokens before each CoT step, which further increases the computation capacity. To the best of our knowledge, no existing work proposes to add new tokens that can be generated by the LMs at inference time. The learned specialization of planning tokens within one task, instead of assigning a token for each specific task, is also novel.

**Math Reasoning** Recently, LM-based math reasoning models have shown to be highly effective (Azerbayev et al., 2023; Yang et al., 2023). Recent studies on complex math reasoning problems usually adopt a CoT-based approach (Zhang et al., 2023; Li et al., 2023) that fine-tunes/prompts LLMs to generate reasoning steps before giving the final answer. Most fine-tuning works are highly dependent on the augmented CoT solutions generated by a strong pre-trained LM (Yue et al., 2023; Yuan et al., 2023; Mukherjee et al., 2023; Luo et al., 2023). Our planning token method can fundamentally improve the fine-tuning performance regardless of the present of augmented CoT data. A possible extension of our work is to combine our method with the CoT data augmentation method to boost the performance even more. Our method is especially related to Zhang et al. (2023). They perform CoT fine-

---

[2]We use **mpnet-base** Song et al. (2020), a default model in Sentence-Transformers Reimers & Gurevych (2019).

tuning of GPT2 by first predicting the math operation of each reasoning step at generation time, which is less efficient than our end-to-end method.

## 5 Conclusion

We proposed using planning tokens to gain better control over reasoning with LLMs. Our planning tokens increase the performance over baselines at a minimal parameter cost. We studied our method's efficacy across three different base models and multiple datasets. First, we see a straightforward extension of our framework which could use multiple planning variables at every reasoning step, each planning variable is responsible for deliberating in a different "view" of that particular reasoning step (Wang et al., 2023). Second, future work should go beyond our heuristic inference procedures and learn the inference network, such as to maximize the marginal log-likelihood of the observed data: we could then interpret the overall model as a Sequential VAE (Goyal et al., 2017). Finally, it is meaningful to continue the exploration towards interpretability and explainability of the planning tokens (Khashabi et al., 2021). We believe such research could shed light on prompt searching/optimization studies performed both by humans (Zamfirescu-Pereira et al., 2023) and machines (Shin et al., 2020; Sordoni et al., 2023).

## 6 Ethics Statement

We do not foresee a significant societal impact resulting from our proposed method. In this work, we propose to equip LLMs with planning tokens to obtain better reasoning capabilities. We find that the planning tokens discovered by embedding-based methods (less interpretable, e.g., K-Means and SQ-VAE) consistently outperform those from heuristics-based methods (more interpretable, e.g., Arithmetic). Therefore, an obvious direction for future work is to improve the interpretability of such methods. In addition, while we aim to improve an LLM's reasoning ability, it is possible that the training of the planning tokens could be affected by misinformation and hallucinations introduced by the base LLM. Therefore, we believe that caution needs to be exercised if one extends our work to a setting where humans are directly involved, e.g., in an educational setting.

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

# A Appendix

**A generation example**

Question: Every day, Wendi feeds each of her chickens three cups of mixed chicken feed, containing seeds, mealworms and vegetables to help keep them healthy. She gives the chickens their feed in three separate meals. In the morning, she gives her flock of chickens 15 cups of feed. In the afternoon, she gives her chickens another 25 cups of feed. **How many cups of feed does she need to give her chickens in the final meal of the day** if the size of Wendi's flock is 20 chickens?

| **Vanilla supervised fine-tuning** | **Fine-tuning with arithmetic planning tokens** |
|---|---|
| Every day, Wendi gives her chickens 15 cups of feed in the morning + 25 cups of feed in the afternoon = <<15+25=40>>40 cups of feed. | <prefix> <+> Wendi gives her flock 15 cups of feed in the morning and another 25 cups in the afternoon, for a total of 15+25 = <<15+25=40>>40 cups of feed. |
| If she has 20 chickens and she feeds them 40 cups of feed every day, then each chicken gets 40/20 = <<40/20=2>>2 cups of feed per chicken. | <prefix> <*> If Wendi has 20 chickens, then she needs 20*3 = <<20*3=60>>60 cups of feed to feed her flock. |
| The answer is: 2 *\* Drifting away from the correct reasoning flow* | <prefix> <-> If Wendi has already given her flock 40 cups of feed, then she needs to give her flock 60-40 = <<60-40=20>>20 more cups of feed. |
| | <prefix> <answer> The answer is: 20 |

Figure 6: An example of a question from the GSM8K dataset, along with two chain-of-thoughts solutions generated by LLMs. **Left**: The reasoning chain generated by vanilla supervised fine-tuning LLM drifts away from correct reasoning flow and leads to a wrong answer. **Right**: we use planning tokens to guide an LLM's generation at every reasoning step to encourage the correct reasoning flow. For simplicity, we show the variant of our method which utilizes arithmetic planning tokens: <prefix>, <+>, <*>, <->, <answer> are the planning tokens added to the original vocabulary.

**Training details**

Due to excessive computational requirements, in the experiments using Llama2 as the base model, we rely on low-rank adapters (LoRAs) (Hu et al., 2021) to fine-tune the base LLM. We apply LoRAs to the projection parameters of each MLP block in the base LLM. For all methods, we use a rank of 16, and a dropout of 0.05. To further reduce the memory usage, we load 13B Llama2 in 8 bits at training time. For Phi-1.5, we perform full-model fine-tuning. For our variants with planning tokens, we add the embeddings of the planning tokens to the learnable parameters of the base model, which increases less than 0.001% of the total trainable parameters. We train all models for 10 epochs, with a learning rate of 2e-5 using the AdaFactor optimizer (Shazeer & Stern, 2018) for full fine-tuning, and a learning rate of 2e-4 using the AdamW optimizer (Loshchilov & Hutter, 2017) for parameter efficient tuning.

**Comparison with prompt tuning**

In Table 4, We show that the prompting (Lester et al., 2021) /prefix (Li & Liang, 2021) tuning-based fine-tuning methods are not suitable for complex reasoning problems, compared to other parameter-efficient fine-tuning methods like LORA. The only way to increase the capacity of these methods is to add more tunable tokens to the prompt while a large number of new tokens will significantly increase the inference time and space complexity. Adding a comparable amount of new tokens to Table 1 yields significantly lower performance than our method combined with LORA.

**Planning token distribution**

In Figure 7, we show the frequency of each planning token that appears in the GSM8K test set. Annotation means the planning type predicted by the SQ-VAE. Generation means the planning token generated by the fine-tuned language model. As we can see, the marginal distribution of the planning variable approximately matches between SQ-VAE and the language model, which means fine-tuning the language model with planning tokens can effectively learn to infer the correct planning type from the previous steps.

| LM | Method | GSM8K | MATH | AQUA | Avg |
|---|---|---|---|---|---|
| Llama2 | Prefix | 8.9 | 3.6 | 32.9 | 15.1 |
| (7B) | Prompt | 15.2 | 5.3 | 26.8 | 15.8 |
|  | LORA | 38.2 | 6.5 | 36.6 | 27.1 |
| Llama2 | Prefix | 16.0 | 3.2 | 30.3 | 16.5 |
| (13B) | Prompt | 27.8 | 6.4 | 26.0 | 20.1 |
|  | LORA | 44.6 | 7.2 | 41.3 | 31.0 |

Table 4: Comparison between LORA and soft prompt tuning based method. We use a similar number of tokens for Prefix tuning and Prompt tuning as our methods in Table 1 to keep the inference time complexity the same.

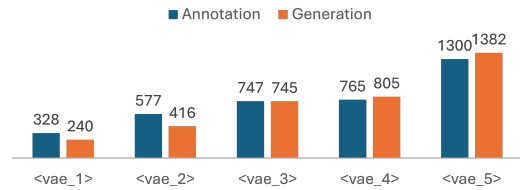

Figure 7: The marginal distribution (count) of the SQ-VAE planning tokens over the GSM8K test set.

**Planning token attention head**

We find the attention head with the largest difference between planning tokens and normal tokens as shown in Figure 8 as the attention head that strongest corresponds to the planning token mechanism.

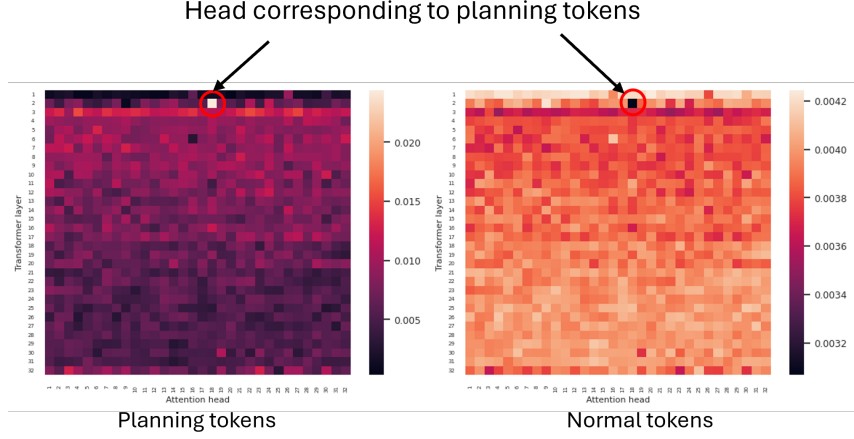

Figure 8: Heat maps of averaged attention on all attention heads of Llama 2 (7B) fine-tuned with SQ-VAE planning tokens on GSM8K.

**GPT4 prompting details for error type classification**

Here is the prompt we used for error type classification:

**Error Classification Prompts**

Your goal is to identify the error type in reasoning steps of an AI agent in response to a question.

You are given the corresponding question, the reasoning steps generated by an AI model and the ground truth reasoning steps in the format:

####Question: [question]

####Generated Answer: [reasoning]

####GT reasoning: [ground truth reasoning].

Your task is to identify the error type in AI's reasoning. There are 5 different error types: (1) Misunderstanding of question, (2) Computation errors, (3) Inaccurate extraction of question information, (4) Wrong application of math knowledge, (5) Wrong logic/unreasonable step.

Here are some examples of each of the error types:

1 Misunderstanding of question: ask about total number of cups in the final meal, answer about number of cups each chicken get.: Question: Every day, Wendi feeds each of her chickens three cups of mixed chicken feed, containing seeds, mealworms and vegetables to help keep them healthy. She gives the chickens their feed in three separate meals. In the morning, she gives her flock of chickens 15 cups of feed. In the afternoon, she gives her chickens another 25 cups of feed. How many cups of feed does she need to give her chickens in the final meal of the day if the size of Wendi's flock is 20 chickens? Every day, Wendi gives her chickens 15 cups of feed in the morning + 25 cups of feed in the afternoon = <<15+25=40>>40 cups of feed. If she has 20 chickens and she feeds them 40 cups of feed every day, then each chicken gets 40/20 = <<40/20=2>>2 cups of feed per chicken. The answer is: 2

2 Computation errors: 7 + 3301x != 3371x: Question: What is the smallest positive integer $N$ such that the value $7 + (30 \times N)$ is not a prime number? We have $7 + (30 \times N) = 7 + 30 + 300 + 3000 + \dots + (30) \times N = 7 + 3301x = 3371x$ We write $3371x$ as $3300+71x$, in order to factor out a multiple of 33 We can try adding one to $71$ and subtracting it from $3300$ If we add 2, we get $73$, but $73$ is prime If we add 4, we get $75$, but $75$ is prime If we add 8, we get $83$, but $83$ is prime If we add 16, we get $99$, but $99$ is prime If we add 32, we get $131$, but $131$ is prime If we add 64, we get $195$, but $195$ is prime If we add 128, we get $323$, but $323$ is prime If we add 256, we get $581$, but $581$ is prime If we add 512, we get $803$, but $803$ is prime If we add 1024, we get $1127$, but $1127$ is prime If we add 2048, we get $1335$, but $1335$ is prime If we add 4096, we get $1544$, but $1544$ is prime If we add 8192, we get $1763$, but $1763$ is prime If we add 16384, we get $2401$, but $2401$ is prime If we add 32768, we get $2777$, but $2777$ is prime Therefore, the smallest integer $N$ is $\boxed{2777}$. The answer is: 2777

---

**Error Classification Prompts (Cont'd)**

3 Inaccurate extraction of question information:
Question: John drives for 3 hours at a speed of 60 mph and then turns around because he realizes he forgot something very important at home. He tries to get home in 4 hours but spends the first 2 hours in standstill traffic. He spends the next half–hour driving at a speed of 30mph, before being able to drive the remaining time of the 4 hours going at 80 mph. How far is he from home at the end of those 4 hours? He was driving at 60mph for 3 hours so he traveled 60*3=<<60*3=180>>180 miles. He was driving for 4 hours and spent 2 hours standing still so he drove 4−2=<<4−2=2>>2 hours. He drove 2 hours at 30mph so he traveled 2*30=<<2*30=60>>60 miles. <– should be half hour He drove 2 hours at 80mph so he traveled 2*80=<<2*80=160>>160 miles. <– should be 1.5 hour He drove 180 miles at 60mph, 60 miles at 30mph and 160 miles at 80mph for a total of 180+60+160=<<180+60+160=300>>300 miles. <– probably because 180−60−160 does not make sense so the model change it to a sum The answer is: 300

4 Wrong application of math knowledge: it's not a geometric sequence: Question: An airplane climbs 100 feet during the first second after takeoff. In each succeeding second it climbs 100 feet more than it climbed during the previous second. How many seconds does it take for the plane to reach an altitude of 12,000 feet above its takeoff height? The number of feet the plane is from the ground is the target of a geometric sequence with first term 0 and common ratio 2 We are asked to find the number of terms of this sequence which have a sum of 12000 The first term is 0, and the common ratio tells us that the second term is 2 more than the first, the third is 2 more than the second, and so on This means that the product of all the terms is $2^0\cdot{0}\cdot{2}\cdot{4}\cdot{8}\cdot{16}\cdot{32}\cdot{64}\cdot{128}\cdot{256}\cdot{512}\cdot{1024}\cdot{2048}\cdot{4096}\cdot{8192}\cdot{16384}\cdot{32768}\cdot{65536}\cdot{131072}\cdot{262144}\cdot{524288}\cdot{1048576}$, and the sum of all these terms is $262144 + 262145 + \cdots + 1048576 = 2\cdot{2^0} + 2\cdot{2^1} + \cdots + 2\cdot{2^{63}} = 2\left(1 + 2 + \cdots + 2^{63}\right)$ $2^{63}$ is equal to $2^{64} − 1$, so we add $1$ to the right hand side to get $2(1 + 2 + \cdots + 2^{64} − 1) = 2\left(2^{64} − 1\right) = 2^{65} − 1$ Finally, we subtract 1 from the right hand side to get $2^{65} − 1 − 1 = 2^{65} − 2^{65} = 1$, so it takes $\boxed{65}$ seconds for the plane to reach an altitude of 12,000 feet. The answer is: 65

5 Wrong logic/unreasonable step: does not make sense to use 6!−4!: Question: In how many ways can we seat 6 people at a round table if John and Sam insist on sitting next to each other? (Two seatings are equivalent if one is a rotation of the other.) The number of all seating arrangements is $6!$ The number of seating arrangements in which John and Sam are next to each other is $4!$ Therefore, there are $6!−4!=\boxed{75}$ different ways to seat the group. The answer is: 75 Question: If $2ab = 12$, evaluate $8a^2b^2$. We have $2ab = 12 = 2^3 \cdot b^2$, so $8a^2b^2 = 2^3 \cdot (8a^2)b^2 = 2^3 \cdot 2b^2 = \boxed{8b^2}$. The answer is: $8b^2$

Format the output as follows: ####Type: [only the index of the error type 1−5] #### Explanation: [short and precise explanation].

Here is an example to analyze:

####Question: ...

####Generated Answer: ...

####GT reasoning: ...

Your response:

