# OpenReview forum: "Guiding Language Model Reasoning with Planning Tokens"
_colmweb.org/COLM/2024/Conference — COLM_

### Official Review · Reviewer_mxHr · 2024-05-10

**Rating:** 7
**Confidence:** 4
**Ethics Flag:** 1

**Summary:**

This paper introduces adding planning tokens in front of each chain-of-thought (CoT) step to guide LLMs on math reasoning. Three different ways to infer the planning tokens are proposed, in which a clustering method and a VAE-based method have shown effective on several math reasoning datasets. The paper is clearly written and the experiments are solid.

**Reasons To Accept:**

The idea of using planning tokens to guide CoT is intuitive, and the proposed K-Means and SQ-VAE -based planning token inference methods have shown a good practice of this idea.

**Reasons To Reject:**

The experiments are limited to finetuning LLMs on a small scope of math reasoning datasets. In the proposed method, the planning tokens should be inferred from a training set using either clustering or generative methods.  The scalability of the method may be a problem when training an LLM with multiple data resources of different distributions,  as it may be exhaustive to obtain the planning tokens for each dataset.

---

> ### Author Rebuttal · Authors · 2024-05-29
>
> We thank the reviewer for their positive review. We would like to respond to the reviewer’s comments below:
>
> Scalability: In this paper, we conduct experiments on small-scale data due to the computation limit. We are interested in scaling our method to the mixture of many different reasoning data and improving the general reasoning ability of language models in the future. Scaling up the number of clusters/planning types and applying our method to the mixture of all datasets might be a solution to this problem.

---

### Official Review · Reviewer_H5H2 · 2024-05-10

**Rating:** 8
**Confidence:** 4
**Ethics Flag:** 1

**Summary:**

This work describes a method for fine-tuning language models to take advantage of additional "planning tokens" injected ahead of reasoning steps when solving problems with multi-step intermediate reasoning.

The authors inject a number of additional embeddings into models' input+output embedding matrices, then augment reasoning data by prefixing each reasoning step in a dataset with some number of planning tokens. The authors evaluate several means of assigning planning token identities when augmenting the training data: using a hand-designed heuristic (the arithmetic operations present in a step), k-means clustering of the step's average hidden states, or quantized variational autoencoder encodings of steps' average hidden states. The LM is then fine-tuned on the resulting data, optionally using a LoRA parameterization.

The authors test their method on three math datasets and observe consistent gains over vanilla fine-tuning. Additional analysis reveals that models do indeed learn to use the planning tokens, and in fact pay more attention to planning token identities when they are generated using k-means or a VAE, indicating that planning token identity is exploited by tuned models.

**Questions To Authors:**

Typos:
The caption for Table 2 contains "We set *n_special = n_special*", presumably this is not intentional.

Suggestions:
In §2.1, page 4, the SQ-VAE equations reference a *softmax* function that goes from latent vectors to the $q$ distribution - this includes a projection from the cluster vector dimension down to P bins, right? Might be good to clarify that this is a little more than the normal "softmax" function.

**Reasons To Accept:**

- The method is intuitively appealing.
- The authors back it up with both consistent improvements in downstream performance numbers and interesting analysis.
- The paper's cleanly written and very easy to follow.
- Points to neat directions for further work.

**Reasons To Reject:**

I don't see any issues that would merit rejection.

The one minor weakness of the work is only evaluating on math reasoning, but that's a pretty common thing to do these days, so it really isn't that much of a problem.

---

> ### Author Rebuttal · Authors · 2024-05-29
>
> We thank the reviewer for their positive review. We would like to respond to the reviewer’s questions below:
>
> Typos: It should be n_prefix = n_special instead of n_special = n_special. Thanks for pointing this out.
>
> Softmax function: Yes, there should be a projection matrix from the neural network hidden size to P bins. We will clarify this in the revision. Thanks for pointing this out.

---

### Official Review · Reviewer_y4dA · 2024-05-11

**Rating:** 6
**Confidence:** 4
**Ethics Flag:** 1

**Summary:**

This paper proposes a Planning token to help generate reasoning steps more effectively in Math Reasoning tasks. The Planning token can be created using three methods (Arithmath, K-Means, SQ-VAE). Among them, the SQ-VAE method, which learns representations more intricately, exhibits the highest performance. While the proposed solution is interesting, it is rather difficult to understand the overall process of the proposed solution.

**Questions To Authors:**

1. How does it construct clusters? Are the clusters also used in the test dataset?

2. Could you show more analysis for clusters?

**Reasons To Accept:**

1. The proposed solution proposes a reasoning path by refining step units.

2. It constructs prompts for error taxonomy and efficiently proceeds with GPT-4 utilization without human labor.

**Reasons To Reject:**

1. Overall, the proposed solution is difficult to understand.

2. It is necessary to show the effect of special tokens in evaluation.

3. It is necessary to compare the baseline solution using pause tokens. Refer to the following paper.

[1] Goyal et al., Think before you speak: Training Language Models With Pause Tokens. CoRR abs/2310.02226 (2023)

---

> ### Author Rebuttal · Authors · 2024-05-29
>
> We thank the reviewer for their positive review. We are happy to see the reviewer finds our work interesting. We would like to respond to the reviewer’s comments and questions below:
>
> Response to reasons to reject:
>
> 1. Writing: We are happy to include any specific modifications that can improve clarity.
>
> 2. Effect of special tokens: Our experiments are conducted to prove the effect of special tokens. In Table 1, we show the effect of different types of special tokens by performance on math reasoning datasets, with and without them. In Figure 2, we show the effect of special tokens on long reasoning. In Figure 3, we show the effect of special tokens on different error types. In Table 3 and Figure 4, we show the effect of special tokens on attention weights.
>
> 3. Pause token baseline: Our General baseline is an upgraded version of the pause token, by essentially adding pause tokens at more positions. To better illustrate this, the accuracy of using the pause token with Llama 2 7B on GSM8K, MATH, and AQUA is 37.2, 6.7, and 36.2. For comparison, the general baseline is 38.5, 6.7, and 37.8, and our SQ-VAE method is 40.0, 7.0, and 41.3. We will add this to the main table for clarity.
>
> Response to questions:
>
> 1. How clusters are constructed: please refer to section 2.1, where we introduce three ways to infer planning token types and construct clusters, including a heuristic way, K-means clustering, and a VAE-base way. The clusters are constructed using training sets. At test time, the cluster/planning type is inferred by the language model.
>
> 2. Cluster analysis: we show a probing-based analysis of clusters in the experiment section. It would be great if the reviewer could elaborate on the possible additional cluster analysis.

---

### Official Review · Reviewer_652L · 2024-05-13

**Rating:** 7
**Confidence:** 4
**Ethics Flag:** 1

**Summary:**

This paper proposes a method to improve LLMs' chain-of-thought reasoning ability on math problems, by adding trainable "planning tokens" at the beginning of each reasoning step. The method is well-motivated, with consistent performance gains demonstrated by solid experimental results. The presentation is clear, which makes the paper easy to follow. However, the experiments involve only one task (math) and three datasets, which can limit the generalizability of the findings. Additionally, the analysis of attention weights may not directly support the causal claim made by the authors.

**Questions To Authors:**

1. The notation "t_i" is overloaded with both "planning variable" and "planning token" and is inconsistent across the paper. In Figure 1, t_i refers to a planning token. In Section 2.1, t_i refers to a planning variable, and "each planning variable can be 'verbalized' by one or more planning tokens", which sounds confusing.
2. Potential typo in Table 2: "we set n_special = n_special" -> should be "n_prefix = n_special"?
3. Confusing sentence in the Conclusion -- “we might want to explore position-wise specialization of the planning tokens, i.e. use a different variant of the planning token w.r.t. the position of the step in the reasoning”: what does "variant" mean?
4. Which model's outputs were used in the error taxonomy analysis?

**Reasons To Accept:**

1. The tackled problem (math reasoning) is important and timely.
2. The method is well-motivated and thoroughly investigated with three ways of implementation, all of which are reasonable. The additional computation cost is minimal when integrated with existing parameter-efficient finetuning approaches (e.g. LoRA).
3. The experimental design is sound, with reasonable baselines and various base LMs.
4. The results show clear and consistent performance gains of the proposed method over baselines across all three datasets.
5. The authors conduct a comprehensive and thoughtful set of analyses to reveal further insights into the strengths and weaknesses of the proposed method.
6. The paper is well-written and easy to follow.

**Reasons To Reject:**

1. The experiments only involve one task (math reasoning) and three datasets, whereas existing works along this line often show applicability to more than one type of reasoning task, such as multi-hop QA, planning, logic inference, etc, in addition to math.
2. The performance gaps are sometimes small for certain settings in Table 1. It would be good to report statistical significance.
3. In Section 3.2 (Analysis), the authors report the performance of both Arithmetic and SQ-VAE in Figure 2, but the text only discusses the latter. The sentence “SQ-VAE outperforms the baseline especially for examples requiring longer reasonings” makes it sound like SQ-VAE is the strongest, but actually Arithmetic performs better on quite a few groups (# reasoning steps =4, 5,8, 9). I’m curious about how to interpret these mixed results.
4. In the analysis of attention on planning tokens, I doubt if the observation “planning tokens are assigned much higher attention weights than normal tokens” can support the claim “planning tokens are in general more helpful for the generation”. There has been a long debate on whether attention weights can be interpreted as token importance faithfully ([[1]](https://arxiv.org/abs/1902.10186), [[2]](https://aclanthology.org/D19-1002/), [[3]](https://aclanthology.org/2020.acl-main.432/), among others). A better alternative can be Attention Flow or Attention Rollout ([[4]](https://aclanthology.org/2020.acl-main.385/)) instead of raw attention weights.

---

> ### Author Rebuttal · Authors · 2024-05-29
>
> We thank the reviewer for their positive review. We would like to respond to the reviewer’s comments and questions below:
>
> Response to reasons to reject:
>
> 1. **Statistical significance**: Thanks for bringing this up. The fine-tuning runs are pretty resource and time-consuming for us so we did not have multiple run results for our main table. To make up for this, we conduct multiple analyses to support our results.
>
> 2. **Reasoning step analysis**: We attribute the mixed performance to improved long reasoning capabilities primarily due to the introduction of specialized new tokens between each reasoning step. While both methods improve on long reasonings, SQ-VAE improves more on average. We will discuss more about this in the revision.
>
> 3. **Attention interpretation**: Thanks for pointing us to the related work. While the raw attention weight might be a debatable way of understanding the token importance, it still serves as a valid way of understanding how the Transformer works (e.g. in recent work like the induction heads [1]). Similar to [1], we also identify attention heads that have strong patterns corresponding to the planning tokens as shown in Figure 4, and deduct how language models make use of the planning tokens from the patterns. We will rewrite the statement that attention weight itself implies token importance.
>
> Response to questions:
>
> 1. **Abuse of notations**: We indeed abuse the notation $t^i$ for both the planning variable and planning token for simplicity, as stated in section 2.  Here $t^i$ can either represent one planning variable, or multiple planning tokens corresponding to one planning variable. In the last sentence of the caption of Figure 1, we explain that $t^i$ can be multiple tokens. We will rewrite the description of $t^i$ to make it clearer.
>
> 2. **Typos**: Thank you for pointing this out. It should be n_prefix = n_special.
>
> 3. **The confusing sentence in the Conclusion**: Here by “variant”, we mean we can assign different planning tokens for different steps, even if they are classified to the same planning variable, to differentiate the position. We will rewrite this sentence to make it clearer.
>
> 4. **Model generating error taxonomy**: It’s Llama 2 7B. We will include this information in the revision.
>
> [1] Olsson, C., Elhage, N., Nanda, N., Joseph, N., DasSarma, N., Henighan, T., ... & Olah, C. (2022). In-context learning and induction heads. arXiv preprint arXiv:2209.11895.

---

> > ### Comment · Reviewer_652L · 2024-06-02
> >
> > Thank you for the response! Is there anything you'd like to say regarding the first point in "reasons to reject"?

---

> > > ### Author Response · Authors · 2024-06-03
> > >
> > > Thanks for your reply! We had a less clear intuition about how planning tokens can help other reasoning types. Maybe for multi-hop QA one can infer a planning variable for each hop. We are working on adding a new dataset, StrategyQA [1], which has a training set with multi-hop questions decomposed into single-hop questions, to see the effect of our proposed method. As math world problems are most aligned with the overall direction we want to pursue in our research, our current paper focuses on them. But it's certainly interesting to see if extensions on other types of reasoning tasks are possible. We will keep you posted on the progress of the new dataset if we can finish before the discussion period.
> > >
> > > [1] Geva, M., Khashabi, D., Segal, E., Khot, T., Roth, D., & Berant, J. (2021). Did aristotle use a laptop? a question answering benchmark with implicit reasoning strategies. Transactions of the Association for Computational Linguistics, 9, 346-361.

---

> > > > ### Comment · Reviewer_652L · 2024-06-06
> > > >
> > > > Sounds good, looking forward to that!

---

### Decision · Program_Chairs · 2024-07-10

**Decision:**

Accept

**Comment:**

This paper proposes an interesting and intuitive method to improve language models' multi-step reasoning ability by introducing "planning tokens" before each reasoning step. While the experiments focus only on math reasoning, which limits generalizability, the promising results and potential for extensions to other reasoning tasks make this a solid contribution. Please make clarifications as recommended by the reviewers and perform any additional analysis of clustering, which would further strengthen the paper. Overall, the reviewers recommend acceptance given the interesting idea, solid experiments, and clear presentation.